# Wheat *TaTIP4;1* Confers Enhanced Tolerance to Drought, Salt and Osmotic Stress in *Arabidopsis* and Rice

**DOI:** 10.3390/ijms23042085

**Published:** 2022-02-14

**Authors:** Yan Wang, Yaqi Zhang, Yinchao An, Jingyuan Wu, Shibin He, Lirong Sun, Fushun Hao

**Affiliations:** State Key Laboratory of Cotton Biology, School of Life Sciences, College of Agriculture, Henan University, Kaifeng 475004, China; wy875643001wy@163.com (Y.W.); 18437905061@163.com (Y.Z.); ayc19980108@163.com (Y.A.); wujingyuan99@163.com (J.W.); sbhe@henu.edu.cn (S.H.)

**Keywords:** wheat *TaTIP4;1*, *Arabidopsis*, rice, seed germination, root growth, drought, salinity, osmotic stress

## Abstract

Tonoplast aquaporins (intrinsic proteins, TIPs) have been indicated to play important roles in plant tolerance to water deficit and salinity. However, the functions of wheat TIPs in response to the stresses are largely unknown. In this study, we observed that transgenic plants overexpressing wheat *TaTIP4;1* in *Arabidopsis* and rice displayed clearly enhanced seed germination and seedling growth under drought, salt and osmotic stress. Compared with wild type plants, *Arabidopsis* and rice overexpression lines had heightened water contents, reduced leaf water loss, lowered levels of Na^+^, Na^+^/K^+^, H_2_O_2_ and malondialdehyde, and improved activities of catalase and/or superoxide dismutase, and increased accumulation of proline under drought, salinity and/or osmotic stresses. Moreover, the expression levels of multiple drought responsive genes clearly elevated upon water dehydration, and the transcription of some salt responsive genes was markedly induced by NaCl treatment in the overexpression lines. Also, the yeast cells containing *TaTIP4;1* showed increased tolerance to NaCl and mannitol, and mutation in one of three serines of *TaTIP4;1* caused decreased tolerance to the two stresses. These results suggest that *TaTIP4;1* serves as an essential positive regulator of seed germination and seedling growth under drought, salt and/or osmotic stress through impacting water relations, ROS balance, the accumulation of Na^+^ and proline, and stimulating the expression of dozens of stress responsive genes in *Arabidopsis* and rice. Phosphorylation may modulate the activity of *TaTIP4;1*.

## 1. Introduction

Drought and high salinity are two crucial environmental stresses that greatly limit plant growth, development and crop productivity worldwide. Drought reduces cellular water contents, and salinity disturbs ion balance. Both stressors can result in hyperosmotic stress. Drought, salt and osmotic stresses also enable the accumulation of excess reactive oxygen species (ROS) in cells, causing oxidative stress in plants [1,2,3]. During evolution, plants have developed complicated regulatory mechanisms to withstand these environmental challenges and the resultant secondary stresses, including the perception and transduction of stress signals, the activation of stress responsive transcription factors, followed by the induction of the expression of a large number of functional genes [3,4,5,6].

Typically, drought signals are sensed and relayed downstream to promote the increase in water uptake from the soil. On the other hand, dehydration stress stimulates the biosynthesis of hormone abscisic acid (ABA) to induce stomatal closure, preventing water loss. Water deficit can also lead to the enhancements in the activities of different antioxidant enzymes like superoxide dismutase (SOD), catalase (CAT), ascorbate peroxidase (APX) and peroxidase (POD) to sequester ROS, and lead to the production of various osmotic stress-protectant metabolites such as proline and trehalose [5,6,7].

Both ABA-dependent and ABA-independent signal regulatory systems have been demonstrated to initiate gene transcription in response to water scarcity in plants. Under dehydration stress, ABA triggers the activation of the key positive regulators, subclass III sucrose nonfermenting (SNF) 1-related kinases (SnRK2s) SnRK2.2, SnRK2.3 and SnRK2.6, which phosphorylate and activate some transcription factors, for instance AREBs/ABFs (ABA-response element (ABRE)-binding proteins/ABRE binding factors), further inducing the expression of numerous stress-responsive genes. In contrast, several transcription factors including dehydration responsive element binding protein 1/C-repeat binding factor (DREB1/CBF) and DREB2 function in ABA-independent manner to increase the transcription of multiple drought-responsive genes under drought stress [5,6,8,9]. To date, many components acting downstream of drought signaling have been identified in plants. They include AtDREB2A, AtDREB1B, AtDREB2B, response to dehydration 29A (AtRD29A)/Low-temperature-responsive protein 78 (AtLTI78) and AtRD29B in *Arabidopsis*, and *OsDREB1*, *OsDREB2*, *OsNAC1*, *OsNAC2* and *r**esponsive to abscisic acid 16* (*OsRAB16C*) in rice [9,10,11,12,13,14].

Like drought stress, salinity stimulates the enhancement of ABA levels in tissues, triggering the ABA relevant responses. Salt stress can also induce the increases in the activities of multiple antioxidases and promote the production of osmotic protectant substances [15]. Moreover, ionic stress, the major challenges imposed by salinity, causes the activation of transcriptional regulatory mechanisms of plants to cope with its adverse effects. Upon saline stress, excess Na^+^ is sensed by the salt receptor, subsequently induces the enhancement of cytosolic Ca^2+^ concentrations in plants. The Ca^2+^ signal is decoded by the Ca^2+^-binding proteins salt overly sensitive 3 (SOS3) and SOS3-like calcium-binding protein8/calcineurin B-like 10 (SCaBP8/CBL10), which activate the serine/threonine protein kinase SOS2. Then, the activated SOS2 phosphorylates and elicits the plasma membrane Na^+^/H^+^ antiporter SOS1 to transport Na^+^ from the cytoplasm to the apoplast. SOS2 and SOS3 also positively modulate the activity of NHX1, a vacuolar Na^+^/H^+^ antiporter. NHX1 transfers Na^+^ from the cytoplasm to the vacuole in plants upon salinity [3,4,6].

Aquaporins (AQPs) are universal water channel proteins that selectively and reversibly assist the movement of water and small neutral solutes across cellular membrane in plants and other organisms [16,17,18]. They regulate plant growth and development through affecting water relations, cell turgor pressure, root water absorption and leaf transpiration, etc. Moreover, AQPs play essential roles in plant response to diverse biotic and abiotic stresses [16,17,18,19]. In higher plants, AQPs can be divided into five subfamilies, the plasma membrane intrinsic proteins (PIPs), tonoplast intrinsic proteins (TIPs), nodulin 26-like intrinsic proteins (NIPs), small basic intrinsic proteins (SIPs) and uncategorized intrinsic proteins (XIPs). Among these, PIPs and TIPs are the dominant players. They mainly contribute to the transport of water across the plasma membrane and tonoplast, respectively. TIPs can be further classified into several subtypes for example TIP1 and TIP2. Each subtype is designated as TIP1;1, TIP1;2, and so on [16,17,19]. There exist 10 TIP members both in *Arabidopsis* and rice. Eleven TIPs have been identified in wheat, the major grain crop in the world [20,21,22].

It has been addressed that some TIPs exert positive effects in plant tolerance to water deficit, salinity and osmotic stress [23,24]. For instance, overexpression of a ginseng *PgTIP1* increases drought tolerance in *Arabidopsis*, and enhances salt stress tolerance in *Arabidopsis* and soybean [25,26,27,28]. Transgenic *Arabidopsis* plants overexpressing *Jatropha curcas TIP1;3* display improved seed germination rates, and promoted seedling growth and seed viability under salt, drought and/or osmotic stresses [29]. Likewise, ectopic expression of *Thellungiella salsuginea TsTIP1;2* in *Arabidopsis* clearly increases plant tolerance to drought, salt and oxidative stresses, and overexpression of tonoplast *TsMIP6* in rice leads to elevated tolerance to salt stress [30,31]. Sade et al. reported that constitutive expression of tomato *SITIP2;2* markedly enhances fruit yield under water stress [32]. Overexpression of *Glycine soja GmTIP2;1*, *GmTIP1;7* and *GmTIP1;8* in soybean also results in noticeable improvement in salt and drought tolerance [33]. However, whether wheat TIPs have roles in plant response to dehydration and salt stress is unclear, and the underlying mechanism remains to be determined. In the present study, we provide evidence that *TaTIP4;1* positively regulates seed germination and seedling growth in *Arabidopsis* and rice under drought, osmotic and salt stress through altering water relations, ROS homoeostasis and Na^+^/K^+^ balance, increasing the levels of proline and inducing the transcription of a multitude of stress responsive genes.

## 2. Results

### 2.1. TaTIP4;1 Expression Was Strongly Induced by PEG, NaCl and H_2_O_2_ in Wheat

TIPs have been addressed to fulfill functions in plant adaption to various abiotic stresses [23]. To gain insight into the roles of *TaTIP4;1* in response to water stress and salinity, changes in its transcript abundances were firstly examined in wheat under stress. Treatment of seedlings with 10% PEG significantly enhanced the transcriptional levels of *TaTIP4;1* at 6, 9, 12 and 24 h in leaves and 6 and 9 h in roots. Likewise, 200 mM NaCl led to clear increases in mRNA abundances of *TaTIP4;1* at all treatment time points in leaves, and 3, 6, 9 and 24 h in roots (Figure 1A,B). Exogenous 100 μM ABA had evident stimulation effects on *TaTIP4;1* expression at 6 and 9 h in roots but had no induced impact in leaves (Figure 1C). Application of 10 mM H_2_O_2_ resulted in pronounced enhancements of *TaTIP4;1* mRNA levels at 3, 6 and 12 h in leaves and 6 and 9 h in roots (Figure 1D). These results indicate that *TaTIP4;1* was markedly induced by drought, salinity, ABA and H_2_O_2_. Tissue expression of *TaTIP4;1* was also measured. It was found that *TaTIP4;1* was strong expressed in roots, stems and leaves (Appendix A).

### 2.2. Generation of Transgenic Plants Overexpressing TaTIP4;1 in Arabidopsis and Rice

To further study the functions of *TaTIP4;1*, we cloned its coding sequence (CDS) and introduced it into *Arabidopsis* and rice, respectively. Several T3 overexpression lines were generated. Arabidopsis line 6 (OE6), 9 (OE9), 10 (OE10) and rice line 1 (OE1), 2 (OE2) and 3 (OE3) showed clearly heightened expression levels of *TaTIP4;1* (Appendix A). Also, the empty vector driven by 35S promoter was introduced into *Arabidopsis*, and transgenic VC (vector control) plants were obtained. As expected, *TaTIP4;1* mRNAs were not detected in VC lines (Appendix A).

### 2.3. Overexpression of TaTIP4;1 Improved Drought Tolerance of Arabidopsis

To determine whether *TaTIP4;1* expression influences plant tolerance to drought stress, differences in the growth of OE6, OE9, OE10, wild type (WT) and VC were compared. Before treatment, the performances of the three OE lines were similar to those of WT and VC (Figure 2A). When water was withheld for another 2 weeks and 4 weeks, the growth of all plants was severely inhibited. Notably, the three overexpressors were observably more tolerant to drought stress than WT and VC. The OE lines still had some green leaves at 2 weeks, and were alive at 4 weeks whereas WT and VC had more dark brown leaves at 2 weeks, and looked dead post 4 weeks of water stress treatment. After rewatering for 1 week following 4-week drought stress, the OE lines survived and resumed growth, but WT and VC plants died (Figure 2A). Moreover, the survival rates and leaf relative water contents were significantly higher in the three OE lines than in WT and VC under drought stress (Figure 2B,C). Additionally, leaf water loss rates of the *TaTIP4;1* overexpression lines were clearly smaller than those of WT and VC plants (Figure 2D). These results suggest that *TaTIP4;1* positively modulates drought tolerance of *Arabidopsis*.

### 2.4. Arabidopsis TaTIP4;1 Overexpressors Had Increased Seed Germination Rates under Salt and Osmotic Stress

To examine whether *TaTIP4;1* plays a role in seed germination under stresses, seeds of OE6, OE9, OE10, WT and VC were sterilized, stratified and sown on solid 1/2 MS medium or 1/2 MS medium supplemented with NaCl (50 and 75 mM) or mannitol (100 and 200 mM), and grown in a chamber for a period of time. In 1/2 MS medium, nearly all the seeds were germinated at 3 d, no significant differences in germination rates among the plants were observed (data not shown). At 10 d, the seed germination rates of the three overexpression lines were noticeably higher than those of WT and VC in 1/2 MS medium containing 50 mM NaCl, 75 mM NaCl, 100 mM manitol or 200 mannitol (Figure 3), hinting that *TaTIP4;1* aids seed germination in *Arabidopsis* after challenge by saline and osmotic stimuli.

### 2.5. Arabidopis Seedlings Overexpressing TaTIP4;1 Had Enhanced Tolerance to Salinity and Osmotic Stress

We then investigated the roles of *TaTIP4;1* in seedling growth of *Arabidopsis* under salt and osmotic stress. Five-day old seedlings of OE6, OE9, OE10, WT and VC were transferred to 1/2 MS medium with or without different concentrations of NaCl or mannitol, and grown in a chamber for another 10 d. In 1/2 MS medium, all plants had similar growth performances. No marked difference in root lengths among the plants was seen. However, in 1/2 MS medium containing 100 mM NaCl, 125 mM NaCl, 300 mM mannitol or 400 mM mannitol, the elongation of primary roots of the three OE lines was observably faster than WT and VC plants (Figure 4).

The responses of *TaTIP4;1* overexpressors to 250 mM NaCl in soil were also studied. As expected, all the three OE lines grew better than WT and VC after treatment with NaCl for another 2 weeks and especially 4 weeks. After treatment for 4 weeks, the OE lines showed more than two times of survival rates of the WT and VC (Appendix A). Together, these data imply that *TaTIP4;1* positively affects *Arabidopsis* tolerance to NaCl and osmotic stress.

### 2.6. Seed Germination of Rice TaTIP4;1 Overexpressors Was Insensitive to Salt and Osmotic Stress

The effects of *TaTIP4;1* expression on seed germination were further investigated in rice under saline and osmotic stress. It was found that seeds of rice OE line 1, 2 and 3 germinated faster than those of WT in the presence of 5% PEG6000, 10% PEG6000, 50 mM NaCl, and 75 mM NaCl. After treatment with PEG6000 or NaCl for 7 d, the three OE lines had pronouncedly higher seed germination rates than WT plants. In contrast, seed germination rates of the OE plants were not significantly different from those of WT under normal growth conditions (Figure 5), indicating that *TaTIP4;1* acts as a positive modulator of rice tolerance to both water deficit and NaCl stress in term of seed germination.

### 2.7. Rice TaTIP4;1 Overexpressors Grew Better Than WT after Challenged by PEG6000, Salt and Mannitol

We next tested whether *TaTIP4;1* overexpression impacts the growth of rice seedlings under stresses. Three old-day seedlings of OE1, OE2, OE3 and WT grown in liquid MS medium were moved to MS medium without or with different concentrations of PEG6000, NaCl or mannitol for a period of time. The three OE lines showed no obvious growth differences from WT in MS medium. However, all the OE lines grew notably faster or better than WT in MS medium supplied with 5% PEG6000, 10% PEG6000, 50 mM NaCl, 75 mM NaCl, 200 mM NaCl or 200 mM mannitol (Figure 6 and Appendix A). The overexpression lines had apparent longer roots and higher shoots under stresses. The heights of OE plants were more than 1.7, 2.0, 1.5 and 1.8 folds in MS medium containing 5% PEG6000, 10% PEG6000, 50 mM NaCl and 75 mM NaCl, respectively (Figure 6). Also, the survival rates of OE1, OE2 and OE3 were 3.6, 3.3 and 3.5 folds of WT, respectively, after grown under 200 mM NaCl for 10 d, and were 2.5, 2.3 and 2.9 times of WT respectively after grown under 200 mM mannitol for 10 d (Appendix A). Furthermore, the leaf water loss of OE1, OE2 and OE3 was clearly slower than that of WT at all time points (Appendix A). These data suggest that TaTIP4;l facilitates rice seedling growth under drought, salt and osmotic stress likely via reducing water loss in tissues.

### 2.8. Salt-Stimulated Increases of Na^+^ Contents Were Markedly Reduced in TaTIP4;1 OE Lines of Arabidopsis and Rice

In order to understand whether *TaTIP4;1* overexpression alters the balance of Na^+^ and K^+^ under salinity, the contents of Na^+^ and K^+^ in *Arabidopsis* OE6, OE9, OE10, WT and VC were compared. Treatment of the seedlings with 250 mM NaCl led to clear increases in the concentrations of Na^+^, and the decreases in the levels of K^+^ in WT, VC and OE lines. Noteworthily, the increments of Na^+^ levels in the OE lines markedly smaller than those in WT and VC whereas the decrements of K^+^ in the OE lines were similar to WT and VC. Thus, the Na^+^/K^+^ ratios of the OE plants were significantly declined compared with WT and VC under salt stress. We also investigated the effects of high salinity on changes in levels of Na^+^ and K^+^ in rice. Similar to *Arabidopsis*, rice OE1, OE2, OE3 and WT plants showed remarkably enhanced contents of Na^+^ and diminished levels of K^+^ after exposure to 200 mM NaCl for 5 d. However, the enhancements in Na^+^ levels were notably smaller in the OE lines than in WT while the diminishments of K concentrations in the OE plants were parallel to WT (Figure 7). Collectively, these results support the notion that *TaTIP4;1* overexpression causes the suppression of Na^+^ accumulation and the maintenance of K^+^ and Na^+^ balance in *Arabidopsis* and rice under saline stress.

### 2.9. Overexpression of TaTIP4;1 Altered H_2_O_2_ Accumulation, MDA Contents, the Activity of Some Antioxidases, and Proline Levels in Arabidopsis upon Drought and Salt Stress

To ascertain whether *TaTIP4;1* expression in *Arabidopsis* modifies the levels of H_2_O_2_ and malondialdehyde (MDA), the activities of some antioxidant enzymes and proline contents, these physiological parameters were further examined. Under normal conditions, the H_2_O_2_ contents of OE6, OE9 and OE10 were similar to WT and VC. Application of drought for 7 d and 200 mM NaCl for 7 d caused considerable increases in H_2_O_2_ levels in all the plants. It is noteworthy that the rise in H_2_O_2_ concentrations in the OE lines was evidently smaller than that in WT and VC after PEG treatment whereas that in the OE lines was similar to WT and VC under salt stress (Figure 8A).

MDA levels reflect the degree of lipid peroxidation in membranes. It was found that MDA contents pronouncedly enhanced in WT and VC rather than in the three OE lines after challenged by 300 mM PEG, and 200 mM NaCl (Figure 8B).

We detected the changes in the activities of CAT, POD and SOD, found that CAT activities substantially increased in the OE plants, and those in the OE lines markedly higher than WT and VC after treatment with 300 mM PEG or 200 mM NaCl (Figure 8C). POD activities marginally elevated and SOD activities did not obviously altered in all plants after exposure to the PEG and NaCl stress, and no differences in these activities were observed among all the plants under normal conditions (Figure 8D,E).

The proline contents of the lines of OE6, OE9, OE10, VC and WT drastically increased under PEG and salt stress but not under unstressed conditions. Moreover, the increments in proline levels of the OE lines were clearly smaller post PEG treatment but higher under NaCl stress than those in WT and VC plants (Figure 8F). Together, these results imply that *TaTIP4;1* overexpression results in the alleviation of oxidative stress possibly through enhancing the activities of CAT, and the improvement of proline levels in *Arabidopsis* under water stress and/or salinity stress.

### 2.10. TaTIP4;1 Expression in Rice Affected the Levels of H_2_O_2_ and MDA, Antioxidase Activities and Proline Accumulation under Salt and Osmotic Stress

In rice, H_2_O_2_ contents in leaves from OE1, OE2 and OE3 were similar to those from WT under normal growth conditions. Treatment with 200 mM NaCl did not cause apparent differences in H_2_O_2_ concentrations between the OE lines and WT. However, application of 250 mM mannitol resulted in notable enhancements in H_2_O_2_ levels in WT but not in the three OE lines (Appendix A). MDA levels in WT were remarkably increased, and those in OE lines were slightly heightened upon both NaCl and mannitol stress (Appendix A). SOD activities under 200 mM NaCl, and CAT activities in the presence of 250 mM mannitol were evidently decreased in WT, but marginally changed in the OE lines. POD activities of the OE lines did not noticeably differ from WT under both stress and control conditions (Appendix A). NaCl and mannitol challenges resulted in marked elevation of proline levels in the three OE lines rather than in WT plants (Appendix A). Our findings suggest that the enhancements of *TaTIP4;1* transcripts in rice cause diminished oxidative stress and promoted biosynthesis of proline under salt and osmotic stress.

### 2.11. The Expression of Several Drought Responsive Genes Was Upregulated in TaTIP4;1 OE Lines of Arabidopsis and Rice under Water Stress

To clarify whether *TaTIP4;1* overexpression influences the mRNA abundances of drought related genes in *Arabidopsis* and rice, the expression levels of some stress responsive genes were detected. When WT and OE plants were challenged by water dehydation, the transcription of *AtDREB1B*, *AtRD29A* and *AtRD29B* in *Arabidopsis*, and *OsDREB2*, *OsNAC1*, *OsPIP2;1*, *OsRAB16C* and *OsPOX1* in rice was markedly induced. Of note, the elevated amplitudes of the expression levels of these genes in the OE lines clearly higher than those in WT (Figure 9 and Appendix A), pointing to the important roles of *TaTIP4;1* in promoting the expression of these drought associated genes under water stress.

### 2.12. The Transcript Abundances of Multiple Salt Responsive Genes Increased in TaTIP4;1 OE Lines of Arabidopsis and Rice upon Salinity

NaCl caused changes in the expression levels of multiple salinity relevant genes were analyzed in WT and the OE lines of *Arabidopsis* and rice. Salt stress led to noticeable elevations in mRNA abundances of *AtSOS2*, *AtNHX1*, *AtRD29A*, *AtRD29B*, *AtPKS5* and *AtCAT1* in *Arabidopsis*, and of *OsPIP2;1*, *OsP5CS* and *OsCATB* in rice. Moreover, the increments of gene expression in the OE plants were prominently greater than those in WT in both *Arabidopsis* and rice (Figure 10 and Appendix A). These data indicate that *TaTIP4;1* positively modulates the transcription of these salt response genes after exposure to saline conditions.

### 2.13. Phosphorylation Might Regulate TaTIP4;1 Activity in Adaptation to Salt and Osmotic Stress

The role of *TaTIP4;1* in responding to salt and osmotic stress was further investigated in yeast. Under unstressful conditions, the growth of yeast cells expressing *TaTIP4;1* in YPD + galactose medium was similar to that of the control cells (expressing the pYES2 empty vector). However, the yeast cells containing *TaTIP4;1* in YPD + galactose supplied with NaCl or sorbitol grew observably faster than the control cells (Figure 11), indicating that *TaTIP4;1* is a positive modulator in yeast tolerance to salinity and osmotic stress. When one of three serines at loci S83A, S201A or S207A in *TaTIP4;1* was mutated to alanine, the recombinant yeast cells grew evidently slower than those bearing normal *TaTIP4;1* under salt and sorbitol stress conditions (Figure 11). Our findings suggest that phosphorylation functions in adjusting the activity of *TaTIP4;1* in yeast, further affecting the tolerance to saline and osmotic stresses.

## 3. Discussion

To date, the functions of wheat TIPs in responding to drought, high salinity and osmotic stresses are largely unknown. In this report, we found that *TaTIP4;1* expression in wheat significantly increased by application of PEG, NaCl and H_2_O_2_. *TaTIP4;1* was pronouncedly induced by PEG in leaves whereas it was upregulated by NaCl and ABA in roots (Figure 1). These data indicate that *TaTIP4;1* may play more important roles in leaves through transferring water under dehydation conditions, while *TaTIP4;1* may majorly function in roots in response to salt and ABA signalings in wheat.

It was observed that transgenic plants overexpressing *TaTIP4;1* in *Arabidopsis* and rice exhibited improved drought tolerance, enhanced seed germination rate and seedling growth under salt and osmotic stress (Figure 2, Figure 3, Figure 4, Figure 5, Figure 6, Appendix A). Compared with WT, Arabidopsis and rice transgenic lines had increased water contents, diminished leaf water loss, lowered levels of Na^+^, Na^+^/K^+^, H_2_O_2_ and MDA, and improved activities of CAT and/or SOD, and increased concentrations of proline under stresses (Figure 2, Figure 7, Figure 8, Appendix A). Furthermore, the mRNA abundances of multiple drought responsive genes clearly increased under water deficit, and those of some salt responsive genes were upregulated by NaCl treatment (Figure 9, Figure 10, Appendix A). These results suggest that *TaTIP4;1* serves as an essential positive regulator of seed germination and seedling growth under drought, salt and/or osmotic stress through impacting water relations, ROS balance, the accumulation of Na^+^ and proline, and eliciting the expression of dozens of stress responsive genes in *Arabidopsis* and rice.

It has been documented that overexpression of *PgTIP1*, *JcTIP1;3*, *TsTIP1;2*, *TsMIP6*, *SlTIP2;2*, *GmTIP2;1*, *GmTIP1;7* and *GmTIP1;8* markedly improves plant tolerance to drought, salinity and/or osmotic stress as indicated by increased seed germination and/or superior growth status [23,25,26,27,28,29,30,31,32,33,34], being in agreement with our findings (Figure 2, Figure 3, Figure 4, Figure 5, Figure 6, Appendix A). These results support the notion that TIPs play more general roles in plant response to drought, salt and osmotic stresses through facilitating water movement across the tonoplast. Peng et al. reported that overexpression of *ginseng PgTIP1* results in reduced water deficit tolerance after grown in shallow (10 cm deep) pots [25]. Feng described that the mutant of Arabidopsis *AtTIP2;2* has increased seedling growth upon drought, salinity and osmotic stresses [35]. These data are inconsistent with our results. The reason may be that wheat *TaTIP4;1* has opposite effects from PgTIP1 and AtTIP2;2 on the direction of water transport across the tonoplast, or the regulatory mechanism of *TaTIP4;1* is diverse from that of PgTIP1 and AtTIP2;2 under stress conditions.

After challenged by dehydration and/or salt stress, the decrements in leaf water contents and/or increments in leaf water loss rates of transgenic plants overexpressing *PgTIP1*, *JcTIP1;3* or *TsTIP1;2* are evidently arrested compared with those of WT plants [27,28,29], being in accordance with our findings (Figure 2 and Appendix A). These results imply that TIPs are of great importance in plant response to drought and salt stress via modulating water transportation.

We observed that salinity-induced increases in the Na^+^ contents and Na^+^ to K^+^ ratios in *TaTIP4;1* overexpressors of *Arabidopsis* and rice were notably lower than those in WT (Figure 7). The results were consistent with those in transgenic plants overexpression of *PgTIP1* in soybean [27,28], signifying that TIPs exert effects in ameliorating ion stress in plants upon saline conditions.

It has addressed that drought- and/or NaCl stress-triggered enhancements in levels of ROS and MDA in the overexpressors of *PgTIP1*, *TsMIP6* and *TsTIP1;2* are attenuated as compared with those in WT [26,28,30,31]. Moreover, high concentrations of NaCl caused increments in the activities of SOD, CAT, APX and POD are clearly higher in the overexpressors of *PgTIP1* than in their corresponding WT plants [27,28]. Likewise, overexpression of *TaTIP4;1* in *Arabidopsis* markedly suppressed the increase in the synthesis of H_2_O_2_ and heightened CAT activity under salt stress (Figure 8). These findings suggest that *TIPs* are essential for the regulation of redox balance in plants upon stress.

Sun et al. demonstrated that salt stress-stimulated increases in proline contents are pronouncedly higher in *TsMIP6* transgenic plants of rice than in WT, being similar to our results from *TaTIP4;1* overexpressors of *Arabidopsis* and rice under drought and salt stress (Figure 8 and Appendix A) [31]. These data hint that some TIPs modulate stress response through stimulating the production of proline.

Numerous lines of evidence indicate that the expression of many stress responsive genes is induced by drought and salinity stress in WT, and especially in the transgenic plants overexpressing *PgTIP1* in *Arabidopsis* and soybean. These genes include *Arabidopsis AtDREB1A*/*AtCBF3*/*AtDREB1A*, *zinc finger of Arabidopsis thaliana* (*AtZAT12*), *AtMYB15*, *cold-regulated 47* (*AtCOR47*), *inducer of CBF expression 1* (*AtICE1*), *AtLTI78*, soybean *GmPOD*, *GmAPX1*, *GmSOS1*, *clathrin light chains* (*GmCLC1*), *GmNHX1*, *GmSOS1*, *GmCAT1*, *9-cis-epoxycarotenoid dioxygenase 1* (*GmNCED1*) and *Δ^1^-pyrroline-5-carboxylate synthetase 1* (*GmP5CS1*) [26,27,28]. We also found that the expression levels of a number of stress related genes like *AtDREB1B*, *AtRD29A* and *AtRD29B* in Arabidopsis *TaTIP4;1* overexpressors, and *OsDREB1*, *OsDREB2*, *OsNAC1*, *OsNAC2*, *OsPIP2;1*, *late embryogenesis abundant 3* (*OsLEA3*), *OsRAB16C*, *catalase B* (*OsCATB*) and *peroxidase 1* (*OsPOX1*) in rice *TaTIP4;1* overexpressors were observably higher than those of WT plants under drought stress (Figure 9 and Appendix A). The mRNA abundances of the salt-associated genes such as *AtSOS2*, *AtNHX1*, *AtRD19*, *AtRD29A*, *AtRD29B*, *AtPKS5* (*SOS2-like protein kinase 5*) and *AtCAT1* in Arabidopsis *TaTIP4;1* overexpressors, and *OsPIP1;1*, *OsPIP2;1*, *OsCATB* in rice *TaTIP4;1* overexpressors were observably higher than those of WT plants under salt stress (Figure 10 and Appendix A). These results suggest that overexpression of *TIPs* significantly impacts the transcript abundances of the genes related to drought, ion transport, ABA signaling, ROS sequestration, water transportation and/or the production of osmotic protectant substances, further improving stress tolerance of plants.

In sum, *TaTIP4;1* overexpression in *Arabidopsis* and rice imparts the tolerance to drought, salt and osmotic stress in terms of seed germination and seedling growth through maintaining the water balance and ROS homeostasis, attenuating the accumulation of Na^+^, promoting the biosynthesis of proline, and inducing the expression of many stress responsive genes. Thus, the TIP can act as a good candidate for genetic manipulation of crop tolerance to water deficit and salinity stress in the future.

## 4. Materials and Methods

### 4.1. Analysis of TaTIP4;1 Expression in Wheat under Stresses

The seeds of wheat (*T. aestivum* L. cv. Chinese Spring) were placed in a plate, germinated and cultured in fresh liquid 1/2 MS medium in a growth chamber (12 h light/12 h dark cycle at 22 °C with light intensity of about 200 μmol m^−2^ s^−1^). Two-week-old wheat seedlings were transferred to liquid 1/2 MS medium containing 20% PEG 6000, 200 mM NaCl, 200 mM mannitol, 10 mM H_2_O_2_ or 100 μM ABA for 0, 1, 3, 6, 9, 12 and 24 h, respectively. The leaf and root samples were then harvested, frozen in liquid nitrogen and stored at −80 °C.

The samples were ground to a fine powder in liquid nitrogen using a mortar and pestle. Total RNA was generated from the powder applying a RNA extraction kit (Vazyme Biotech, Nangjing, China). The cDNA was synthesized from the RNA by M-MLV reserve transcriptase synthesis kit (Promega, Madison, WI, USA). The expression of *TaTIP4;1* (XM_044521132) was detected using qRT-PCR method with the cDNA, SYBR Green Master mix, the specific primers of *TaTIP4;1* and *TaActin* (AB181991.1, the internal control) (Appendix A), and an ABI 7500 real-time PCR system.

### 4.2. Generation of TaTIP4;1 Transgenic Lines of Arabidopsis and Rice

A 768 bp *TaTIP4;1* CDS was amplified by PCR from wheat cDNA using *TaTIP4;1* special primers (Appendix A). The PCR product was cloned into the binary vectors pBI121 (for *Arabidopsis*) and pCAMBIA1300 (for rice), respectively. The two constructs were sequenced and transformed into *Agrobacterium* strain GV3101 (for *Arabidopsis*) and EHA105 (for rice), respectively, and introduced into *Arabidopsis* plants (Columbia, Col-0) by the floral dip method, and into rice (*Oryza sativa* L. spp. Japonica, var Nipponbare) by the agrobacterium-mediated genetic transformation method. The pCAMBIA1391 empty vector was also introduced into *Arabidopsis* flowers. Multiple T3 transgenic plants were obtained. Total RNA and cDNA were produced from the entire plants, and the transcripts of *TaTIP4;1* were assayed by qRT-PCR following the method described above. Arabidopsis gene *AtActin2* and rice gene *OsActin* acted as the internal controls. Their primer sequences were listed in Appendix A.

### 4.3. Performance Identification of Arabidopsis Overexpressors of TaTIP4;1 upon Drought Stress

Two-week-old Arabidopsis *TaTIP4;1* overexpressors and control (WT and VC) seedlings grown in solid 1/2 MS medium were transplanted in the mixed nutrient soil (rich soil:vermiculite = 2:1, *v*/*v*), and grown in a growth chamber (21/18 °C day/night temperature cycle, 16/8 h light/dark cycle with the light intensity of ~120 μmol m^−2^ s^−1^) for another two weeks. Then, the plants were not irrigated for next two weeks and four weeks, and then rewatered for one week. The survival rates were scored after rewatering. Leaf water loss rates were measured at room temperature.

### 4.4. Analysis of Seed Germination of Arabidopsis and Rice under Stresses

Seeds of Arabidopsis WT, VC and *TaTIP4;1* overexpression lines were sterilized, washed, and sown on solid 1/2 MS medium (containing 0.8% agar and 3% sucrose) with or without 50 mM NaCl, 75 mM NaCl, 100 mM mannitol or 200 mM mannitol. After 3 days of stratification at 4 °C, the seeds were germinated in a growth chamber (21/18 °C day/night temperature cycle, 16/8 h light/dark cycle with light intensity of ~120 μmol m^−2^ s^−1^) for indicated periods of time. The WT and *TaTIP4;1* overexpressing rice seeds were disinfected, washed and cultured in liquid MS medium with or without 5% PEG6000, 10% PEG6000, 50 NaCl or 75 mM NaCl for periods of time in the Arabidopsis growth chamber described above. Photographs were taken and seed germination rates were calculated.

### 4.5. Analysis of Seedlings Growth of Arabidopsis and Rice upon Stresses

For *Arabidopsis* plants grown in medium, 5-day-old WT, VC and *TaTIP4;1* overexpression seedlings grown vertically on solid 1/2 MS solid medium were transferred to 1/2 MS medium supplied with 100 NaCl, 125 mM NaCl, 300 mM mannitol or 400 mM mannitol for indicated periods of time. The length of primary roots was measured and photographs were taken. For *Arabidopsis* plants grown in soil, 10-day-old WT, VC and *TaTIP4;1* overexpression seedlings were transferred to nutrient soil mentioned above for another 2 weeks. Then, the plants were watered with 250 mM NaCl or deionized water (the control) every 2 d for next 2 weeks or 4 weeks. The survival rates were determined according to the bleaching of leaves on the fourth day. For rice plants, 3-day-old WT and transgenic seedlings grown in liquid 1/2 MS medium were treated with or without 5% PEG6000, 10% PEG6000, 50 mM NaCl or 75 mM NaCl for another one week. Then, the plant height of all lines was measured. Additionally, 20-day-old WT and transgenic seedlings grown in liquid 1/2 MS medium were treated with or without 200 mM NaCl or 200 mM mannitol for another 10 d. Survival rates and water loss rates of the plants were then examined.

### 4.6. Measurement of Na^+^ and K^+^ Contents under Salt Stress

For *Arabidopsis*, 30 day-old WT, VC and *TaTIP4;1*-overexpressor plants grown in soil were watered with deionized water or 250 mM NaCl for another 2 weeks. The entire rosettes were collected. About 0.1 g of fresh sample was put in a pre-frozen mortar and pestle, ground with 0.5 mL deionized water on ice, and centrifuged (4 °C, 3500 rpm) for 10 min. The supernatant was used to analyze Na^+^ and K^+^ contents applying the kits C002 and C001 (Nanjing Jiancheng Bioengineering Institute in China, www.njjcbio.com, accessed on 26 December 2021), respectively, following their instructions. For rice, 20-day-old seedlings grown in liquid 1/2 MS medium were treated with or without 200 mM NaCl for 2 weeks. Leaves were harvested, and Na^+^ and K^+^ contents were measured following the method above.

### 4.7. Detection of the Levels of H_2_O_2_ and MDA, the Activities of Antioxidases and Proline Accumulation upon Stresses

For *Arabidopsis*, 21-day-old WT, VC and *TaTIP4;1* overexpression plants grown in soil in the *Arabidopsis* growth chamber described above were watered with deionized water or 200 mM NaCl once every two days for 7 d. The entire rosettes were sampled for further analysis. For rice, 21-day-old WT and *TaTIP4;1* overexpressor plants grown in liquid 1/2 MS medium were treated without or with 200 mM NaCl or 250 mM mannitol for 5 d. The leaves of the plants were harvested. About 0.2 g fresh *Arabidopsis* and rice samples were placed in the pre-frozen mortar and pestle, and ground with 1.8 mL phosphate buffered solution (0.1 mM, pH 7.4) on ice. The homogenate was centrifuged (4 °C, 8000 rpm) for 10 min. The supernatant was applied to monitor the contents of H_2_O_2_, MDA and proline, and the activities of CAT, POD and SOD by their corresponding detection kits (A064 for H_2_O_2_, A003 for MDA, A107 for proline, A007 for CAT, A084 for POD, and A001 for SOD). The kits were purchased from Nanjing Jiancheng Bioengineering Institute of China (www.njjcbio.com, accessed on 26 December 2021).

### 4.8. Gene Expression Analysis by qRT-PCR

In Arabidopsis, the expression levels of *AtRD29A* (AT5G52310), *AtRD29B* (AT5G52300), *AtRD19* (AT4G39090), *AtDREB1A* (AT4G25480), *AtDREB2A* (AT5G05410) and *AtDREB2B* (AT3G11020) after PEG treatment, and those of *AtSOS1* (AT2G01980), *AtSOS1* (AT2G01980), *AtSOS2* (AT5G35410), *AtSOS3* (AT5G24270), *AtMOCA1* (AT5G18480), *AtPKS5* (AT2G30360), *AtCAT1* (AT1G20630) and *AtNHX1* (AT5G27150) under salt stress were examined. Likewise, rice mRNA abundances of *OsDREB1A* (AGC24705.1), *OsDREB2A* (JQ341059), *OsPIP1;1* (AK061769), *OsPIP2;1* (AK072519), *OsNAC1* (BAC53810.1), *OsNAC2* (BAC53811.1), *OsLEA3* (CAA92106.1), *OsRAB16C* (BAT13917.1), *OsCATB* (BAA05494.1) and *OsPOX1* (LOC_Os01g15830) upon PEG challenge, and of *OsSOS1* (AWX66759.1), *OsNHX1* (BAA83337.1), *OsNAC6* (AB098712), *OsTIP4;1* (AAS98488.1), *OsP5CS1* (BAA19916.1) and *OsHKT1* (AFY08290.1) under salt stress were determined. For *Arabidopsis*, 15-day-old WT, VC and *TaTIP4;1* transgenic plants grown in solid 1/2 MS medium were treated with or without 300 mM PEG6000 or 200 mM NaCl for 6 h. The entire rosettes were collected. For rice, 30-day-old WT and transgenic plants overexpressing *TaTIP4;1* grown in liquid 1/2 MS medium were treated without or with 300 mM PEG6000 or 200 mM NaCl for 12 h. The leaves of the plants were sampled. qRT-PCR experiments were carried out using the Arabidopsis and rice samples according to the methods described above. Genes *AtActin2* and *OsActin* were used as reference genes in *Arabidopsis* and rice, respectively. The sequences of all used primers were listed in Appendix A.

### 4.9. Analysis of TaTIP4;1 Functions in Yeast

The protein structure of *TaTIP4;1* was predicted using the online software (https://services.healthtech.dtu.dk/service.php?TMHMM-2.0, accessed on 26 December 2021). *TaTIP4;1* is a six transmembrane protein. Three potential phosphorylation sites (S83, S201 and S201) in the cytoplasmic region were selected for further analysis. The *TaTIP4;1* CDS and the CDS with mutated nucleotides (the 247th, 602th, 619th) were amplified, and cloned into yeast expression vector pYES2 driven by a GAL1 promoter, respectively. The constructs were introduced into yeast strain INVSC1 (His-, Leu-, Trp-, and Ura-) following the standard lithium acetate method [36]. For salt stress, the yeast cells were cultured in 5 M NaCl for 24 h at 30 °C with shaking. For sorbitol stress, the yeast cells were cultured in 2 M sorbitol for 24 h at 30 °C with shaking. Then, the yeasts were grown on selective solid plates containing 50 mM NaCl or 200 mM sorbitol at 30 °C for 3 days [37].

### 4.10. Statistical Analysis

At least three independent experiments were conducted. All the qRT-PCR experiments technically repeated nine times. Data are means ± SD. Statistical analyses were performed by one way ANOVA and Tukey’s HSD test (*p* < 0.05) or student’s *t*-test to examine data differences.

## Figures and Tables

**Figure 1 ijms-23-02085-f001:**
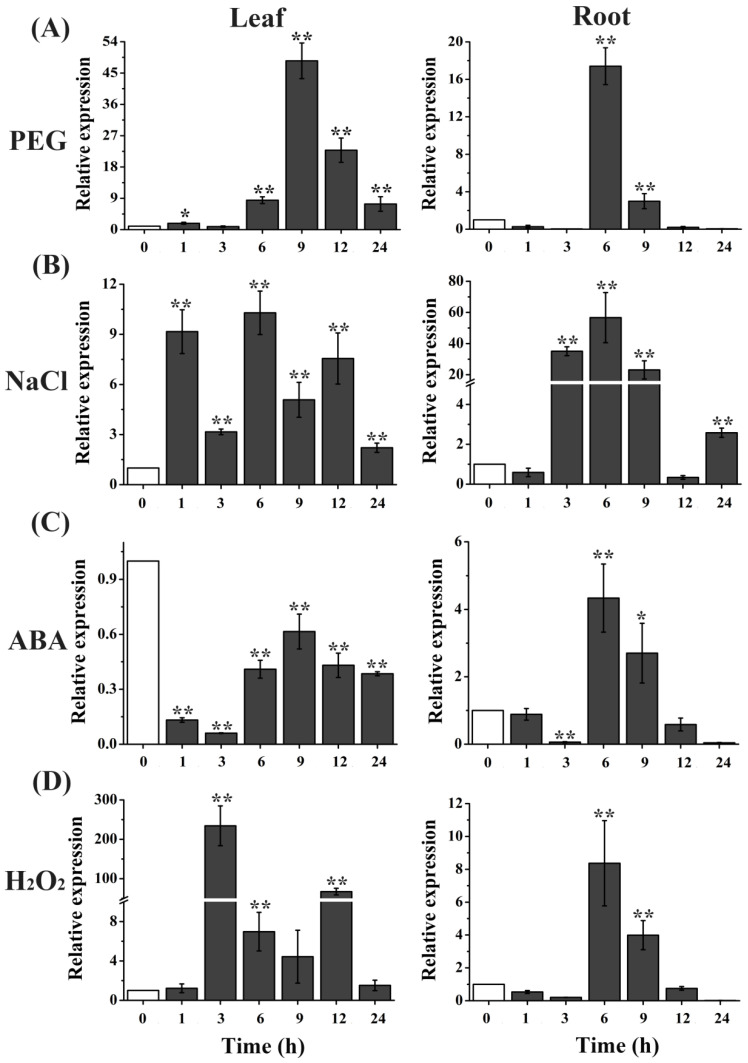
Expression profiles of *TaTIP4;1* after treatment with PEG6000, NaCl, ABA and H_2_O_2_ in wheat. Two-week-old wheat seedlings were treated with 10% PEG 6000 (**A**), 200 mM NaCl (**B**), 100 μM ABA (**C**) or 10 mM H_2_O_2_ (**D**) for indicated periods of time. qPCR analysis of *TaTIP4;1* transcription was performed. *TaActin* was used as an internal control. Data are mean ± SD (n ≥ 3). Single and double asterisk mean that the data from the treatment samples significantly differed from the control at * *p* < 0.05 and ** *p* < 0.01 levels respectively by student’s *t* test.

**Figure 2 ijms-23-02085-f002:**
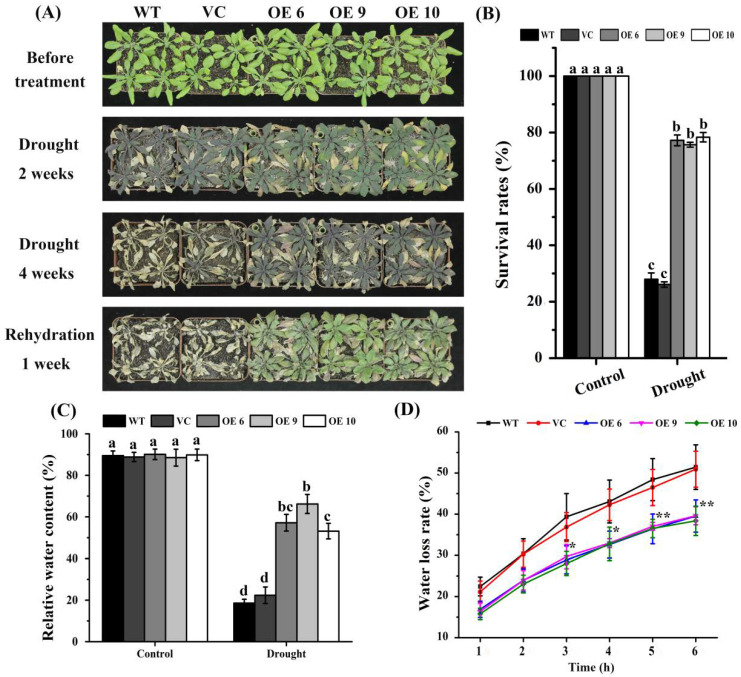
*TaTIP4;1* overexpression improved drought tolerance in *Arabidopsis*. (**A**) Growth performances of plants. Four-week-old plants of Arabidopsis WT, VC, OE6, OE9 and OE10 were not watered for 2 and 4 weeks, respectively, and then rewatered for 1 week. (**B**) The survival rates of plants after drought stress for 4 weeks. (**C**) Relative water contents of leaves after drought stress for 4 weeks. Different lowercase letters show that the values from the plants significantly differed from each other by one way ANOVA and Tukey’s honestly significant difference test (Tukey’s HSD test) (*p* < 0.05). (**D**) Leaf water loss rates of four-week-old plants. Single and double asterisk indicate that the values of *TaTIP4;1* overexpressors significantly differed from WT and VC plants at * *p* < 0.05 and ** *p* < 0.01 levels respectively by student’s *t* test. Data are mean ± SD (n ≥ 3).

**Figure 3 ijms-23-02085-f003:**
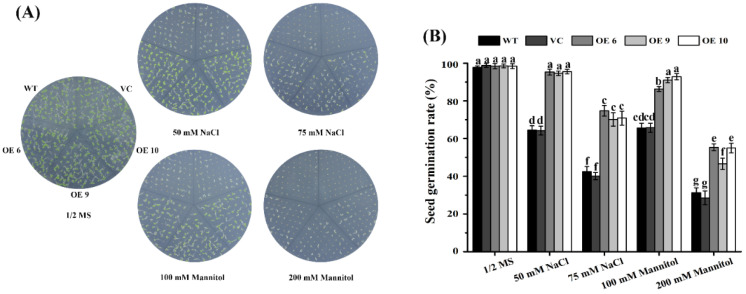
Effects of NaCl and mannitol on seed germination of Arabidopsis *TaTIP4;1* overexpressors. (**A**) Phenotypes of seed germination and seedling growth of various plants. OE6, OE9, OE10, WT and VC were germinated in 1/2 MS medium not containing or containing 50 mM NaCl, 100 mM NaCl, 100 mM mannitol or 200 mM mannitol for 10 day. (**B**) Seed germination rates of various plants in (**A**). Data are mean ± SD (n ≥ 3). Different lowercase letters above the error bars reveal the data of various plants significantly differed from each other by one way ANOVA and Tukey’s HSD test (*p* < 0.05).

**Figure 4 ijms-23-02085-f004:**
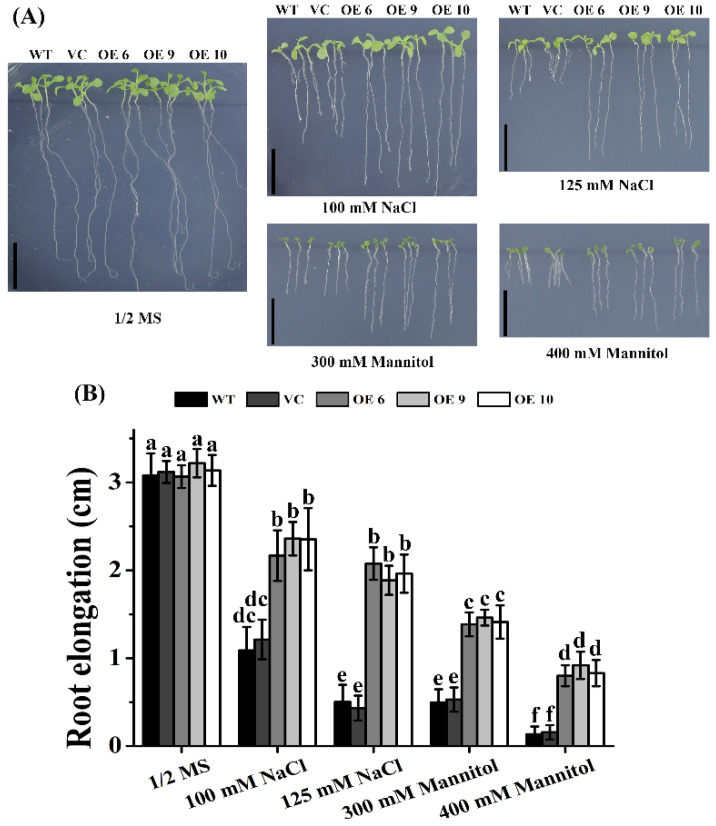
Effects of NaCl and mannitol on primary root growth of Arabidopsis transgenic plants overexpressing *TaTIP4;1*. (**A**) Seedling growth performances. Five-day old seedlings of OE6, OE9, OE10, WT and VC were transferred to 1/2 MS medium supplemented without or with 100 mM NaCl, 125 mM NaCl, 300 mM mannitol or 400 mM mannitol, and grown for another 10 days. Scale bars are 1 cm. (**B**) Increments in the length of primary roots of various lines in (**A**). Values are mean ± SD (n ≥ 3). Different lowercase letters above the error bars indicate that the values of the lines significantly differed from each other by one way ANOVA and Tukey’s HSD test (*p* < 0.05).

**Figure 5 ijms-23-02085-f005:**
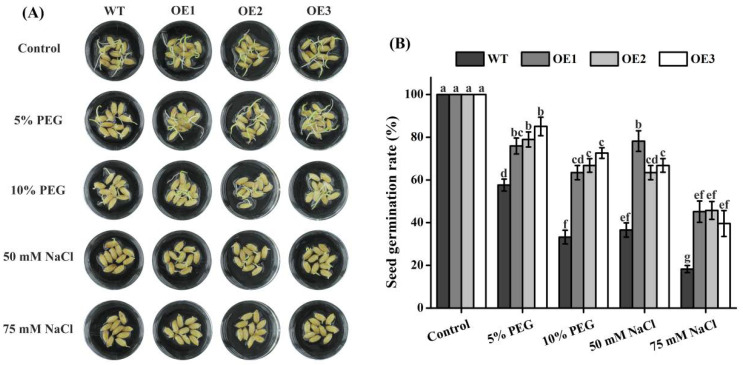
Effects of PEG6000 and NaCl on seed germination of rice WT and *TaTIP4;1* overexpressors. (**A**) Seed germination performances. Seeds of WT, OE1, OE2 and OE3 were germinated in the absence or presence of 5% PEG6000, 10% PEG6000, 50 mM NaCl or 75 mM NaCl for 7 days. (**B**) Seed germination rates of all lines in (**A**). Data are mean ± SD (n ≥ 3). Distinct lowercase letters above the error bars show that the data of various plants significantly differed from each other by one way ANOVA and Tukey’s HSD test (*p* < 0.05).

**Figure 6 ijms-23-02085-f006:**
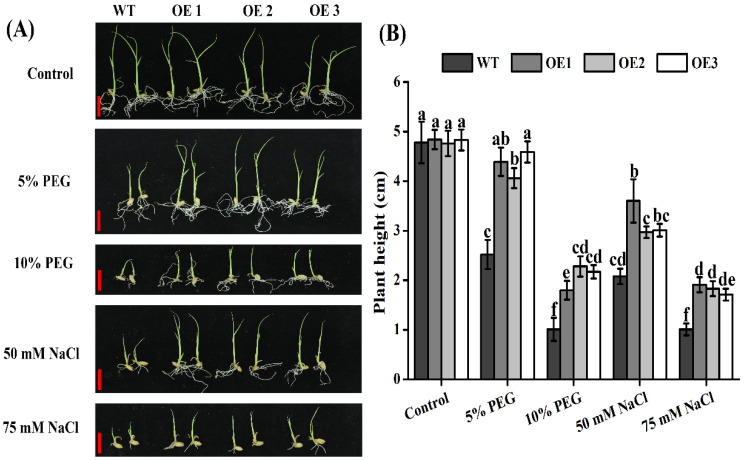
Effects of PEG6000 and salt on the growth of rice WT and transgenic lines. (**A**) Seedling growth performances. Three-day-old seedlings of WT and transgenic lines grown in liquid 1/2 MS medium were treated without or with 5% PEG6000, 10% PEG6000, 50 mM NaCl or 75 mM NaCl for another one week. Scale bars are 1 cm. (**B**) Plant height of all plants in (**A**). Data are mean ± SD (n ≥ 3). Diverse lowercase letters above the error bars represent that the height values of various plants were significantly different from each other by one way ANOVA and Tukey’s HSD test (*p* < 0.05).

**Figure 7 ijms-23-02085-f007:**
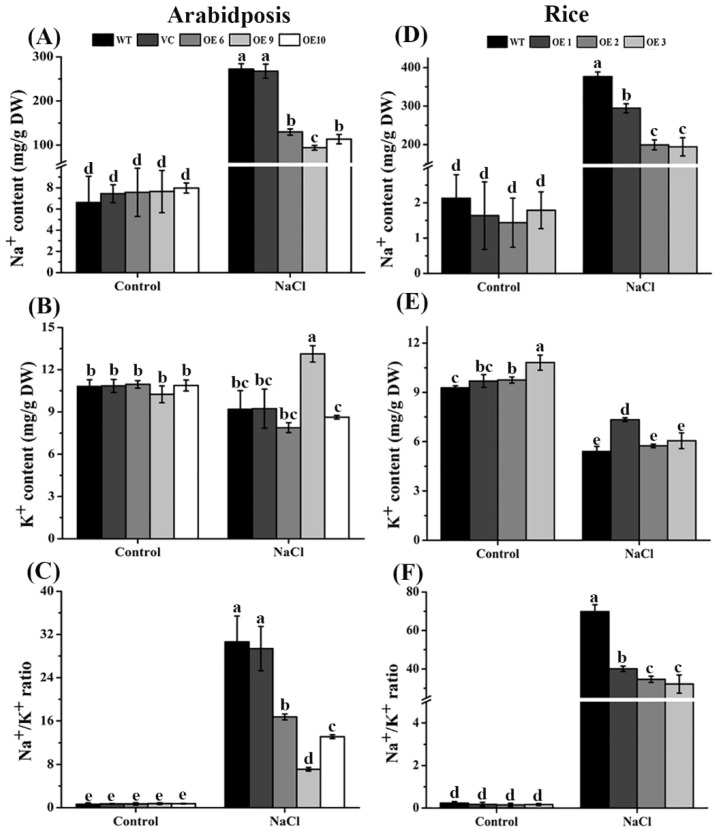
Effects of salt on the accumulation of Na^+^ and K^+^ in WT and transgenic plants of *Arabidopsis* and rice. WT and various OE plants were exposed to NaCl for 2 weeks, and leaves of the plants were collected for measuring the contents of Na^+^ and K^+^. (**A**–**C**) were Na^+^ contents, K^+^ contents and Na^+^/K^+^ values in *Arabidopsis*, respectively; and (**D**–**F**) were Na^+^ contents, K^+^ contents and Na^+^/K^+^ values in rice, respectively. Values are mean ± SD (n ≥ 3). Different lowercase letters above the error bars indicate that the data of various plants significantly differed from each other by one way ANOVA and Tukey’s HSD test (*p* < 0.05).

**Figure 8 ijms-23-02085-f008:**
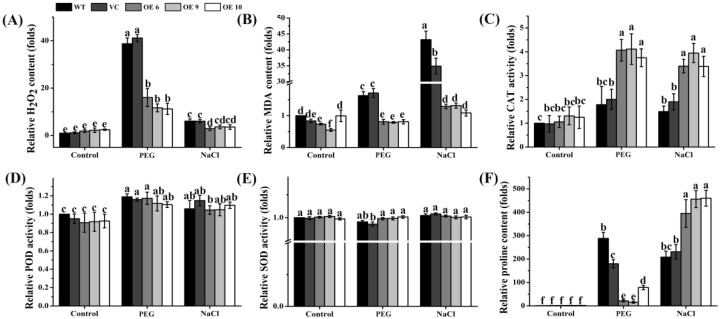
Effects of PEG6000 and NaCl treatment on the levels of H_2_O_2_ and MDA, the activity of some antioxidases, and proline contents in *Arabidopsis* overexpressors of *TaTIP4;1*. (**A**) H_2_O_2_ levels. (**B**) MDA levels. (**C**–**E**) The activities of CAT, POD and SOD, respectively. (**F**) Proline levels. Fifteen-day-old WT, VC and *TaTIP4;1* overexpression seedlings were treated with water or 200 mM NaCl for 7 d. Levels of H_2_O_2_ and MDA, the activity of some antioxidases, and proline contents were assayed. Data are mean ± SD (n ≥ 3). Distinct lowercase letters above the error bars reveal that the values of various plants significantly differed from each other by one way ANOVA and Tukey’s HSD test (*p* < 0.05).

**Figure 9 ijms-23-02085-f009:**
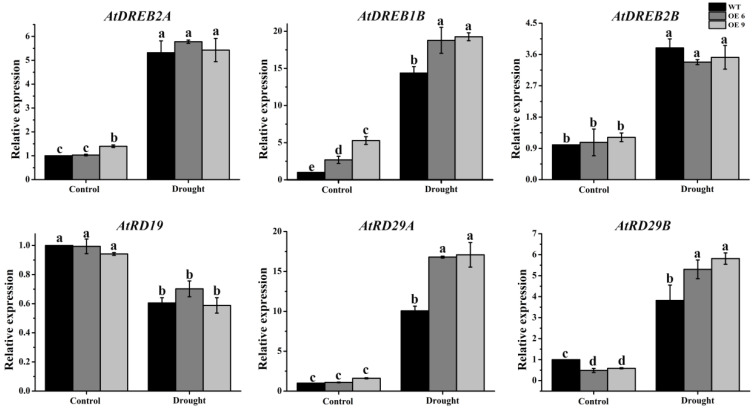
Effects of water deficit on the transcription changes of multiple drought responsive genes in Arabidopsis *TaTIP4;1* overexpressors. Fifteen-day-old WT, VC and *TaTIP4;1* transgenic plants were dehydrated for 6 h. Gene expression was detected by quantitative real-time RT-PCR (qRT-PCR) method, *AtActin2* was used as the internal control. Data are mean ± SD (n ≥ 3). Diverse lowercase letters above the error bars show significant differences in the values among the plants by one way ANOVA and Tukey’s HSD test (*p* < 0.05).

**Figure 10 ijms-23-02085-f010:**
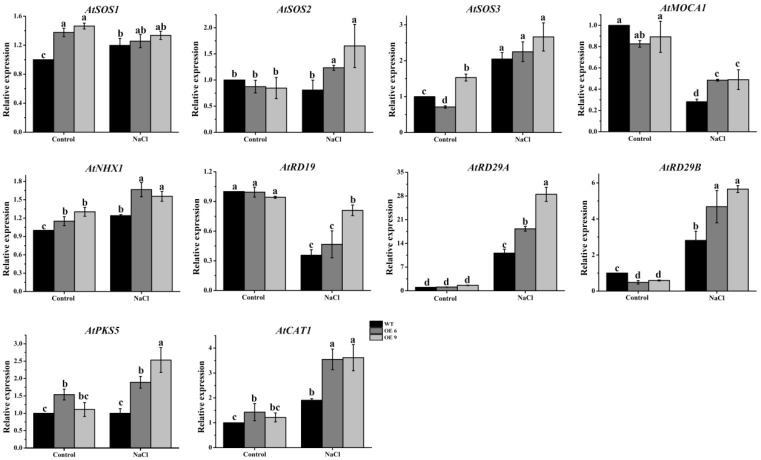
Effects of NaCl on the expression patterns of multiple salt responsive genes in *Arabidopsis* transgenic plants overexpressing *TaTIP4;1.* Fifteen-day-old WT, OE6 and OE9 seedlings were treated with or without 200 mM NaCl for 6 h. Gene expression was evaluated by qRT-PCR method, *AtActin2* was used as the internal control. Values are mean ± SD (n ≥ 3). Diverse lowercase letters above the error bars reveal remarkable differences in the values among the seedlings by one way ANOVA and Tukey’s HSD test (*p* < 0.05).

**Figure 11 ijms-23-02085-f011:**
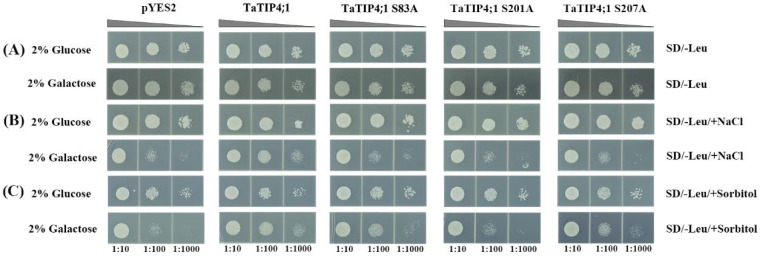
The roles of *TaTIP4;1* in yeast. Growth performances of yeast cells of INVSC1 containing normal or mutated *TaTIP4;1* genes (at the 247th, 602th, 619th, respectively) in medium (SD/-Leu) supplied without (**A**) or with 50 mM NaCl (**B**) and 200 mM sorbitol (**C**) for 3 days at 30 °C. The reduced cell densities in the dilution series are shown by narrowing triangles when proceeding from left to right. Experiments were repeated three times.

## Data Availability

Not applicable.

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
