# Peer review of "Wheat TaTIP4;1 Confers Enhanced Tolerance to Drought, Salt and Osmotic Stress in Arabidopsis and Rice"

_ijms, 2022, doi:10.3390/ijms23042085_

Round 1

Reviewer 1 Report

The manuscript is overall well written and conclusions are supported by the data presented.  However, I provided few comments and request authors to address them to bring more clarity to the current manuscript.

  1. Authors should check the ABA effect on seed germination and seedling growth. An ABA sensitivity assay could help authors identify whether TIP4 functions via ABA-dependent or independent drought signaling. The reduced water loss suggests that TIP overexpressors are ABA sensitive.
  2. The expression level of TIP4 in roots and leaves under control conditions will be helpful to know the tissue-specific expression of TIP4.
  3. ABA and NaCl had a pronounced effect on TIP4 expression in roots, whereas PEG in leaves. At least in the discussion, the author needs to provide some insight into this. 
  4. Supplemental data does not have any figures or data. Please upload a new file with the figures. 
  5. Figure 4. The author transferred 5-day old seedlings to MS plates, NaCl, and mannitol plates and measured root length over the following ten days. It seems the seedlings shown in the 1/2 MS plates are smaller for the 15-day growth. What was the root length of 5-day old seedlings? Also, seedlings look too small for five-day growth, especially in the 400 mM M mannitol. The author should provide a 5-day old seedling picture and the root length. It would be nice to measure root elongation after transferring five-day-old seedlings to corresponding plates instead of total root length. Moreover, there are no scale bars provided for the images.  
  6. What is the basis for considering S81, 201, and 201 as potential phosphosites? 
  7. The method for detecting H202, MDA, and proline accumulation levels should be detailed. The Arabidopsis plants were grown on MS medium for 21 days and then treated for seven days with or without water to collect samples to detect proline and others. It is, however, confusing why plants were grown in MS medium for 21 days. Did the author grow them on plates? And how exactly were plants treated with or without water for seven days? 
  8. The activities of CAT POD and SOD were measured using detection kits. Please provide the kit name and necessary details so that reader can find details. 
  9. Figure legends were poorly written and presented. The author should direct readers to each panel in the figure legends. Every figure legend should mention the method used for statistical significance, whether one-way ANOVA was used or t-test? 

Author Response

Question 1: Authors should check the ABA effect on seed germination and seedling growth. An ABA sensitivity assay could help authors identify whether TIP4 functions via ABA-dependent or independent drought signaling. The reduced water loss suggests that TIP overexpressors are ABA sensitive.

Answer 1: Thanks for the reviewer’s informative suggestions. In this study, we attempted to uncover the roles of TaTIP4;1 in response to drought, salt and osmotic stress, not considering the roles of TaTIP4;1 in ABA signaling. At present, it is very hard for us to finish these experiments described above in a short time period. 

Question 2: The expression level of TIP4 in roots and leaves under control conditions will be helpful to know the tissue-specific expression of TIP4.

Answer 2: Thanks for the good advice. Tissue-specific expression of TaTIP4;1 was analyzed using RT-PCR methods. The results were added in the revised manuscript (Please see lines 123-124, Figure S1).

Question 3: ABA and NaCl had a pronounced effect on TIP4 expression in roots, whereas PEG in leaves. At least in the discussion, the author needs to provide some insight into this.

Answer 3: Thanks for the remindings. These results were discussed in the revised file (Please see lines 387-390).

Question 4: Supplemental data does not have any figures or data. Please upload a new file with the figures.

Answer 4: We are so sorry for the mistake. All the figures were included in the supplemental materials this time.

Question 5: Figure 4. The author transferred 5-day old seedlings to MS plates, NaCl, and mannitol plates and measured root length over the following ten days. It seems the seedlings shown in the 1/2 MS plates are smaller for the 15-day growth. What was the root length of 5-day old seedlings? Also, seedlings look too small for five-day growth, especially in the 400 mM mannitol. The author should provide a 5-day old seedling picture and the root length. It would be nice to measure root elongation after transferring five-day-old seedlings to corresponding plates instead of total root length. Moreover, there are no scale bars provided for the images.

Answer 5: The comments are reasonable. Indeed, the seeds in the experiments for figure 4 were stored for longer time, and germinated slower. Similar seedlings (5-day-old) with roots of about 0.5 cm were selected and transferred to 1/2 MS plates containing NaCl or mannitol. The increment in root elongation was measured, and scale bars were provided for the images (Please see Figure 4).

Question 6: What was the basis for considering S83, 201, and 207 as potential phosphosites?

Answer 6: There are multiple phosphorylation sites in TaTIP4;1. Of these are S83, S201, and S207. The three sites are localized in the cytoplasmic region. We speculated that TaTIP4;1 in the cytoplasmic region may play more important roles, and the three sites may be potential phosphosites. Related information was added in Materials and methods (Please see the lines 573-576).

Question 7: The method for detecting H202, MDA, and proline accumulation levels should be detailed. The Arabidopsis plants were grown on MS medium for 21 days and then treated for seven days with or without water to collect samples to detect proline and others. It is, however, confusing why plants were grown in MS medium for 21 days. Did the author grow them on plates? And how exactly were plants treated with or without water for seven days?

Answer 7: Sorry for the unclear description. Detailed information were provided in the revised manuscript (Please see lines 541-543).

Question 8: The activities of CAT POD and SOD were measured using detection kits. Please provide the kit name and necessary details so that reader can find details.

Answer 8: Thanks for the good suggestions. Detailed information about the kits were added in the revised manuscript (Please see lines 548-550).

Question 9: Figure legends were poorly written and presented. The author should direct readers to each panel in the figure legends. Every figure legend should mention the method used for statistical significance, whether one-way ANOVA was used or t-test?

Answer 9: Thanks for the helpful suggestions. All of the figure legends were revised following the requirement in the new manuscript.

Reviewer 2 Report

The review on the publication by Wang et al. under the title Wheat TaTIP4;1 confers enhanced tolerance to drought, salt and osmotic stress in Arabidopsis and rice

I wanted to say that the authors demonstrated very detailed work. I have only minor questions what can be corrected during the manuscript publication:

Line 27 Change the word Wheat from bold to the standard font.

Fig3 A, please change the figure; I can barely recognize what you would like to show.

Fig 4 A, 6 A, please add scale bars.

Author Response

Question 1: Line 27 Change the word Wheat from bold to the standard font.

Answer 1: Thanks for the reminding. The revision was made.

Question 2: Fig3 A, please change the figure; I can barely recognize what you would like to show.

Answer 2: Thanks for the reviewer’s good suggestions. Figure 3A was corrected.

Question 3: Fig 4 A, 6 A, please add scale bars. 

Answer 3: Thanks for the reviewer’s comments. The scale bars in Figure 4A and 6A were added.

Round 2

Reviewer 1 Report

The authors addressed most of my comments.  However, I have a few minor suggestions/corrections.  

The Arabidopsis plants were approximately 30 days old (21 +7). So, referring to them as seedlings is inaccurate. The procedure is still unknown in this regard. For the assay, what tissue is being sampled (the entire rosette, particular leaves, or the entire plant)? Were the plants at this point in their reproductive stage? Were these plants grown in a greenhouse or a growth chamber? What growth conditions were used (day length, temperature, and light intensity)? How were the samples collected and processed (in liquid nitrogen and homogenized using a grinder)? This information should be included in the materials and procedures.

The statistical methods employed are now included in the figure legends; however, the number of replicates is still missing (technical or biological).

Author Response

Question 1: The Arabidopsis plants were approximately 30 days old (21 +7). So, referring to them as seedlings is inaccurate. The procedure is still unknown in this regard. For the assay, what tissue is being sampled (the entire rosette, particular leaves, or the entire plant)? Were the plants at this point in their reproductive stage? Were these plants grown in a greenhouse or a growth chamber? What growth conditions were used (day length, temperature, and light intensity)? How were the samples collected and processed (in liquid nitrogen and homogenized using a grinder)? This information should be included in the materials and procedures.

Answer 1: Thanks for the reviewer’s valuable suggestions. Seedlings were corrected to plants. The detailed information was provided in the revised manuscript (Please see lines 475, 480-481, 503-504, 516-517, 533, 537-544, 548-556, 575). 

Question 2: The statistical methods employed are now included in the figure legends; however, the number of replicates is still missing (technical or biological).

Answer 2: Thanks for the advice. The number of replicates were described in “4.10. Statistical Analysis”. Detailed information was added in all figure legends and 4.10 (Please see lines 595-596) in the revised manuscript and supplimental materials.